# COVID-19 Pandemic Affects the Medical Students’ Learning Process and Assaults Their Psychological Wellbeing

**DOI:** 10.3390/ijerph18115792

**Published:** 2021-05-28

**Authors:** Marium Aftab, Alsaleem Mohammed Abadi, Shamsun Nahar, Razia Aftab Ahmed, Syed Esam Mahmood, Manik Madaan, Ausaf Ahmad

**Affiliations:** 1Dow University of Health Sciences, Karachi 74200, Pakistan; marium_aftab_1993@yahoo.com; 2Department of Family and Community Medicine, College of Medicine, King Khalid University, Guraiger, Abha 6252, Saudi Arabia; mabade@kku.edu.sa (A.M.A.); sklil@kku.edu.sa (S.N.); raftab@kku.edu.sa (R.A.A.); 3KIMS Medical College Bangalore, Bengaluru 560004, India; manik5009@gmail.com; 4Department of Community Medicine, IIMSR, Integral University, Kursi Road, Lucknow, Uttar Pradesh 226026, India; ausaf.ahmad86@gmail.com

**Keywords:** impact, psychological wellbeing, learning, COVID-19 pandemic, medical students

## Abstract

Background: With the emergence of the COVID-19 pandemic, people are living within a milieu of stress, anxiety, and fear. Medical students are susceptible to these emotional injuries, but their psychological wellbeing and learning may further be assaulted by future uncertainties and altered teaching and training programs. Our objective was to find the extent of the psychological impact of the pandemic and the learning difficulties they are experiencing; Methodology: This cross-sectional study included 418 undergraduate and postgraduate medical students from all over the world. A questionnaire was uploaded in Google survey form. It included background characteristics, questions for psychiatric impact like PHQ-9, GAD-7, ZF-OCS, and questions for learning difficulties perceived in comparison to the pre-pandemic time. Results: Among participants, 34.9% of students were male and 65.1% female. Around 46.4% belonged to the WHO, Eastern Mediterranean region, 26.8% from South East Asia region, 17.5% from the region of America, 5.5% from the European region,2.2%from the Western Pacific region, and 1.7% from the African region. Symptoms due to psychiatric illness were noticed in 393 (93.1%); depression in 386 (92.3%), anxiety in 158 (37.8%), obsessive compulsion disorder in 225 (53.8%), and post-traumatic stress syndrome in 129 (39.9%). Female gender, geographical region, and history of previous psychiatric illness were significantly related to almost all the psychiatric illnesses. Regarding learning difficulty, 96% of students faced problems: trouble with memorizing in 54.0%, concentration problems in 67.0%, about 55.5% of students made more mistakes, while 44.5% noted an increase in reaction time for solving questions. In addition, 90% experienced greater difficulty in overall learning during the pandemic in comparison to the pre-pandemic time. Conclusion: Assault on psychological wellbeing, struggling to memorize, inattention and difficulty in concentration on studies, along with perceived overall trouble with learning, have emerged as collateral damage from the COVID-19 pandemic with respect to medical students.

## 1. Introduction

As the horizon of the year 2020 extended, the world experienced the traumatic crisis of the Corona pandemic, commonly known by the name COVID-19 pandemic, that not only increased morbidity and mortality due to this deadly infection but also the fear of contracting the disease, mitigation efforts (e.g., travel restrictions, stay-at-home orders, school closures, social distancing, isolation), economic strain, and uncertainty about the future associated with it, which have all had an intense indirect impact on social, financial and psychological repercussions in the population globally.

Historically pandemics have always been associated with chaos, fear, isolation, and social breakdown that evoke psychological problems. Hypervigilance can arise because of fear and anxiety and result in post-traumatic stress disorder (PTSD) and/or depression [1]. In Hong Kong, about 70% of people expressed anxiety about getting SARS when the SARS outbreak occurred [2], and a moderately high (median 6/9) degree of anxiety during the A/H1N1 influenza pandemic [3].

During the recent pandemic of COVID-19 infection globally, the general population and the frontline healthcare workers also became vulnerable to the emotional impact of the crisis; this has been reflected by increased reports of domestic violence [4] and many psychological problems, including frustration, stress, anxiety, and depression [5]. In a survey from the United States in April 2020 that involved 1468 respondents from the general population, 13.6% of adults reported serious psychological distress, compared with 3.9% in 2018, at levels that predict serious mental illness [6]. The psychological distress was especially observed to be more in health care workers. For example, a survey involving a total of 1257 participants in the United States revealed symptoms of depression in 50.4%, anxiety in 44.6%, insomnia in 34.0%, and distress in 71.5% [7]. In comparison, a large survey from health care workers in China documented the prevalence rate of traumatic stress to be present in 73.4%, depression 50.7%, generalized anxiety 44.7%, and insomnia in 36.1% of the participants [8].

Medical students being closely associated with the frontline health care workers, are susceptible to experience the same emotional trauma augmented further by future uncertainties and altered teaching and training programs. Globally, most undergraduate and postgraduate medical students have had their clinical training almost blocked [9], doctors in training have had rotations modified or cancelled to maximize the capacity of healthcare systems to cope with pressure from cases of COVID-19. Along with disturbance in clinical training, the classroom teachings and laboratories have been altered and replaced by distant online lessons, leaving them to continue their studies remotely [10,11]. As a result; they are facing challenging circumstances to continue their studies that make them even more vulnerable in getting psychological disorders.

Hence, the entire cohorts of medical students and doctors in training are at risk of entering into a vicious cycle of stress, learning difficulties, further stress, etc. In the face of this new reality, and to limit the damages of the current COVID-19 pandemic on the future frontline workers, it is essential to figure out the extent of psychological impact and learning difficulties they are experiencing. This may help in formulating policies and strategies to break this vicious cycle and support the medical students’ wellbeing through adaptive flexibility for curriculum innovation, culturally sensitive resilience, and wellbeing interventions.

Therefore, the present study aims to find out the extent of the impact of the COVID-19 infection pandemic on the learning and psychology of medical students all over the world and delineate some of the factors that could influence the vulnerability or resistance to such impact.

### 1.1. Primary Objectives

To observe the effect of the COVID 19 pandemic on psychological wellbeing among medical students both at undergraduate and postgraduate levels globally.

To observe the influence of the COVID 19 pandemic on learning difficulties perceived by medical students, both at undergraduate and postgraduate levels globally.

### 1.2. Secondary Objectives

To co-relate the socio-demographic factors with the impact of the COVID 19 pandemic in the case of a positive result.

To relate the learning difficulties with the psychological illnesses if found among the undergraduate and postgraduate medical students.

## 2. Methodology

Study Design: This is a cross-sectional descriptive study

Study participants and Sampling Technique: The study included undergraduate and postgraduate students studying medicine all over the world. The study tool was uploaded in the Google survey form and was posted to multiple medical undergraduate and postgraduate social media groups, including the International Federation of Medical Students’ Associations (IFMSA). The responding students were then grouped according to the WHO regions, i.e., AFRO, PAHO, SEARO, EURO, EMRO, WPRO.

Sample size: The sample size was calculated using the Roasoft, Inc. U.S. online sample size calculator, on the assumption that the total population of approachable medical students maybe 20,000, the prevalence of the impact of the pandemic as 50% (as it is not known from previous studies), 95% confidence interval and 5% acceptable errors, the sample calculated was 384, this sample was increased to 400 to compensate for partial-response. As 418 students responded we included them all.

Study Tool: A questionnaire was developed having three major parts:

Background characteristics included; age, gender, country, COVID-19 cases and mortality rate reported in the country, their level of studies, study environment, employment, financial support, and presence of any previous psychiatric disease diagnosed.

The instruments to reveal psychiatric impact are validated screening questions that included: PHQ-9: Screening Instrument for Depression [12] which has demonstrated 61 percent sensitivity and 94 percent specificity for mood disorders in adults, and 89.5 percent sensitivity and 77.5 percent specificity in adolescents. GAD-7 Scales for Anxiety screening [13], a cutoff score of 8 points demonstrates strong sensitivity (92 percent) and specificity (76 percent) for the diagnosis of GAD.

Screening tool for Obsessive compulsion disorder (OCD), (Zohar–Fineberg Obsessive-Compulsive Screen (ZF-OCS)) the information for this is taken from Obsessive-Compulsive Disorder. London, UK: NICE; 2005 [14].

Screening Instrument for Post-Traumatic Stress Disorder (PTSD) [15] which demonstrated excellent diagnostic accuracy (AUC = 0.941; 95% C.I.: 0.912–0.969) in previous study.

Learning and studying difficulties in comparison to the pre-pandemic time were assessed with a questionnaire having eight items. The tool was developed according to the need from a validated study on this subject [16].

### Scoring

The questionnaire used to assess the psychological wellbeing was rated on a four-point scale (0–3) in which; 0 = Not at all, 1 = several days, 2 = more than one-half the days, and 3 = nearly every day.

For depression, the interpretation was as follows: 1–4 minimal, 5–9 mild, 10–14 moderate, 15–19 moderately severe, and 20–27 was said to be severe.

A positive GAD-7 result was a score of at least 8 points.

For OCD and PTSD, scoring of at least three positive answers was considered as significant.

Only the instrument used for depression shows the severity grades, whiles the validated screening tools for the others can only show the chance of having the disorder and needs proper diagnostic evaluation for the grading of severity.

Regarding the impact on learning, each item was graded individually on the two-point scale as 1. = No and 2. =Yes. Frequency and percentage were applied to describe variables, and the chi-square test was used to find the relation of this impact to psychological illnesses.

Inclusion Criteria: The study included the following:

Undergraduate medical students in the clinical phase of their studies.

Postgraduate students who are preparing actively and planning to appear in any qualification equivalence exams like PLAPS, USMLE or those preparing for higher qualifications such as Board exams, membership, or fellowship exams.

There was no restriction for age or months after graduation as medical studies can be at any age and can be planned any time after graduation.

Exclusion Criteria; All incomplete responses and students from other field and medical students still studying in the basic phase.

Statistical analysis was conducted using the SPSS version 16.0 SPSS Inc. Chicago, U.S. All tests were carried out at the 5% level of significance. Chi-square test was used to test the association between categorical variable. Fisher exact test was applied whenever indicated. Multivariate analysis was performed using the binary logistic regression model to determine the OR for a student diagnosed with any psychiatric disorder and related factors such as age, gender, country, level of study, study enrollment, study environment, corona positive, corona mortality, GAD, PHQ, OCD, and PTSD.

## 3. Results

### 3.1. Back Ground Characteristics 

A total of 418 undergraduate and postgraduate medical students from all over the world answered the survey questionnaire. There were 146 (34.9%) male and 272 (65.1%) female participants. Among these, 194 (46.4%) students who responded belonged to the Eastern Mediterranean (EMRO) region of the WHO, while 112 (26.8%) from the South-East Asia Region (SEARO), 73 (17.5%) students from the Region of America (PAHO), 23 (5.5%) from the European region (EURO), 9 (2.2%) from the Western Pacific region (WPRO) and 7 (1.7%) from the African (AFRO) region. Most of the students were studying undergraduate medicine (*n* = 292, 69.9%), and the rest were postgraduate medical students; including residents for the board, fellowship, and those preparing for equivalence certification exams like USMLE and PLAP (*n* = 126, 30.1%).

The majority of students belonged to regions that had greater than 500 Corona positive cases (*n* = 309, 73.9%) and more than a 100 Corona mortality count (*n* = 257, 61.5%) while the remaining belonged to regions with less than 500 Corona positive cases (*n* = 309, 73.9%) and less than a 100 Corona mortality count (*n* = 116, 38.5%). A small number of subjects had a history of a previous psychiatric illness (*n* = 71, 17%), whereas the majority did not present a history of any previous psychiatric illness (*n* = 347, 83%). A large proportion of the students were unemployed (*n* = 316, 75.6%) while only some of them were employed (*n* = 102, 24.4 %). The financial support was from the government for 48 (11.5%) students, from parents for 312 (74.6%), and 58 (13.9%) students managed to fulfill their financial need by themselves. More students were enrolled with an institute (*n* = 234, 56.0%), while 184 (44%) students were not enrolled with an institute. Studying at home was preferred by the majority (*n* = 365, 87.3%) and fewer studied outside the home in other places like the library, café etc. (*n* = 53, 12.7%) (Table 1). 

### 3.2. Psychiatric Illness

Except for 29 (6.9%) students, the majority 389 (93.1%) had some symptoms due to oneof the common psychiatric illnesses namely; depression, anxiety, obsession compulsion disorder (OCD), or post-traumatic stress disorder (PTSD). Among them, most of the medical students (*n* = 386, 92.3%) showed some degree of depression in contrast to the small number who had no symptoms related to depression (*n* = 32, 7.7%). Anxiety-related symptoms were present in 158 (37.8%), while 260 (62.2%) denied any symptom of anxiety. The presence of symptoms indicating further evaluation of OCD were found in 225 (53.8%) participants, whereas 193 (46.2%) had no complaint related to OCD. A small number of participants had symptoms related to PTSD (*n* = 129, 39.9%), and the majority had no PTSD related symptoms (*n* = 289, 69.1%) Table 2.

The different degrees of severity of depression seen in the participants were minimal and mild (*n* = 226, 58.50%), moderate and moderately severe (*n* = 136, 35.20%), and severe (*n* = 24, 6.3%) Figure 1.

A high prevalence of psychiatric illness was noticed in the present pandemic era of 389 (93.1%) in comparison to students who had a history of psychiatric illness in the pre-pandemic time period (*n* = 71, 17%). A hundred percent of those participants who had a history of previous psychiatric illness developed the symptoms at the present time, and 318 participants developed psychiatric illness including; depression, anxiety, OCD, and PTSD. Table 3 shows the relation between students previously suffering from psychiatric illness with psychiatric illness at the present time.

It was found that out of 389 students, who had a psychiatric illness, the frequency of thenumber of disorders (depression, anxiety, OCD and PTSD) was as follows: only one of these illnesses was present in 124 (29.7%) students, any of two disordersin106 (25.4%), three of them in 74 (17.7%),and all four illnesses were present in 85 (20.3%) students.

Table 4 shows the chi-square test results of the students diagnosed with any psychiatric disorder and the background characteristics that show a significant relation.

For students with depression; gender (*p*-value = 0.003), geographical region (*p*-value = 0.032), Corona mortality count (*p*-value = 0.017), employment (*p*-value ≤ 0.001), and history of previous psychiatric illness (*p*-value = 0.008) were found to be significantly related while age, Corona cases count, level of studies, financial support, enrolment status, and study environment did not show any significant relation; *p*-value more than 0.05.

For students with anxiety, only two demographic factors were significantly related that included the geographical region (*p*-value = 0.04) and the history of previous psychiatric illness (*p*-value ≤ 0.001), whereas all other factors had no significant relation.

Similarly, for participants showing the presence of OCD the same two factors were significantly related namely geographical region (*p*-value = 0.01) and history of previous psychiatric illness (*p*-value = 0.04)

For those with PTSD, gender (*p*-value = 0.007) and history of previous psychiatric illness (*p*-value = 0.001) were found to be significantly related.

### 3.3. Learning Difficulties (as Perceived by the Students When Compared to the Pre-Pandemic Era)

The majority of participants revealed that they had some difficulty in studying (*n* = 400, 95.7%) while a small number of students said they did not notice any learning difficulty compared to the pre-pandemic era (*n* = 18, 4.3%). When asked whether the present COVID-19 pandemic had any negative effect on their memory capacity, a large proportion of participants agreed that they had this learning difficulty (*n* = 216, 54.0%). While a smaller proportion said, there was no effect on memorizing (*n* = 184, 46.0%). Similarly a greater number of students had difficulty to focus/concentrate on studies (*n* = 268 67.0%) whereas 132 (33.0%) were not affected. With regards to the question inquiring about the increase in reaction time for solving questions; a fair proportion of the students felt that they took more time in solving a question for learning in their course 222 (55.5%) although 178(44.5%) students had not noticed such an increase in reaction time. Many students, 214 (53.5%) thought they were making more mistakes when they solved questions in contrast to 168 (46.5%) who did not experience such a change during this period with the COVID-19 pandemic. The overall majority of 360 (90.0%) did feel that it is more difficult to study and learn during this period in comparison to the pre-pandemic time—Table 5 shows further details.

Figure 2 shows the severity of these learning difficulties in the form of a number of positive responses to questions on enlightening the learning problems related to; memory, concentration, reaction time (mind alertness), number of mistakes, study timings, adaptation to changing study environments (as implicated during this pandemic), redirecting attention away from negative thoughts, and the overall learning difficulties they face. Among the background factors only gender appeared to be significantly related to these learning difficulties, female students (*n* = 267, 66.8%) facing significantly more problems than males (*n* = 133, 33.25%); with *p*-value = 0.001. On the other hand, as documented in Table 6 the presence of psychiatric illnesses shows a significant association with learning difficulties perceived by the students when tested by chi-square test except for PTSD showing an insignificant relation with depression (*p*-value = 0.174).

Table 6 shows that the multivariate analysis was performed using the binary logistic regression model to determine the OR for a student diagnosed with any psychiatric disorder and related factors. Age, gender, education level etc., were incorporated into the model for a psychiatric disorder. The Hosmer–Lemeshow test was used to determine the goodness of fit of the model, with a *p*-value > 0.05 demonstrating that the model fit the data. The multivariate analysis revealed that GAD-7 scales for anxiety screening and corona mortality were associated with a diagnosis of psychiatric disorder, as students from GAD-7 anxiety scales and corona mortality had a 2.50-fold and 2.57-fold greater chance of exhibiting psychiatric disorders respectively.

## 4. Discussion

With the emergence of COVID-19 pandemic in 2020, almost complete lifestyle changes occurred; this indirectly led to mental stress, psychological influences, and psychiatric disorders at various levels of society. This has been documented in many studies from individual countries and also in occasional reviews [6,7,8,17,18].Our study was conducted to find out the prevalence of common psychological disorders and learning difficulties perceived specifically among medical students during this era of the pandemic; from all over the world. After extensive literature search, there was no study found related to learning difficulties as perceived by medical students during this COVID-19 pandemic. While there are few studies throwing light on the psychological impact of the COVID-19 pandemic on medical students. [18,19,20,21,22,23].

The study shows a very high proportion (93.1%) of undergraduate and postgraduate medical students having some degree of psychological problem while only 17% of them had any history of psychiatric illness in the pre-pandemic period; this assault on psychiatric wellbeing during the pandemic period is in accord with studies on medical students in Pakistan [19], UK [20], Japan [21], India [22], Morocco [23] and nursing students in Indonesia [18]. Also, several studies reflect increased stress, anxiety, and depression as an influence of the COVID-19 epidemic among all strata of society, including the general public [5,6,17], health care workers [3,8], older adults [24] and students of all categories [25], In contrast, a study from Iran [26] shows only 27.6% of medical students having depression. Depression and anxiety did not significantly differ before and after the COVID- 19 outbreak; this difference could be a regional finding due to different tools.

The highest number of students (92.3%) were suffering from depression of various severity, mostly minimum to mild (58.5%) compared to other common psychiatric illnesses, including anxiety, OCD, and PTSD. This finding is similar to a recent study [23] conducted in Morocco using the same measuring tools for depression and anxiety and revealing that among all the common psychiatric illnesses, depression was more common than the others. In their study with a sample of 549 medical students, 410 (74.6%) reported depression while 341 (62.3%) had anxiety, 344 (62.6%) insomnia, and 379 (69%) distress. Similarly, a study from Chennai, India [22] among undergraduate medical students that also included resident interns, found a higher prevalence of depression than other psychiatric illnesses during the COVID-19 outbreak; it revealed that 35.5% had symptoms of depression while 33.2% and 24.9% had anxiety and stress symptoms respectively.

Depression and PTSD were significantly related to the female gender. Such association was shown in multiple studies among medical students and the general population before the pandemic [27] and also after the COVID-19 pandemic [19,23,28]. This may reflect the biological difference in females, as discussed in a research article [29], while a study from India [22] did not show such an association.

Previous history of any psychiatric illness in the past was significantly associated with depression, anxiety, OCD, and PTSD in our study. This could be related to the fact that relapse rates and exacerbation of symptoms of all pre-existing mental health problems have increased during the COVID 19 pandemic as emphasized recently [30,31]; a study was carried out explicitly among people with OCD [32] which showed that patients with OCD had higher scores on all OC symptom dimensions and symptom severity during the pandemic as compared to their scores from before the pandemic. It has been documented also in another study that stress associated with the COVID-19 outbreak rendered students at higher risk for relapse or new episodes of their pre-existing mental disorders [31].

One interesting finding in our study was the significant relation of world geographic regions with psychiatric illnesses, including depression, anxiety and OCD during the COVID-pandemic. This could reflect different cultural values and lifestyles that influence the reaction and resilience of medical students that ensure psychological well-being. To the best of the author’s knowledge, there have been no studies to document such differences, and further research in this area may be beneficial in identifying and coping with factors leading to the assault on psychiatric well-being of the future health providers in times of disaster like the present pandemic.

Our study did not directly relate the COVID-19 cases count and mortality with most of the psychological illnesses such as anxiety, OCD, and PTSD. Only the students reporting symptoms of depression were significantly related to mortality count. This supports the role of unpredictability, uncertainty, the seriousness of the disease, misinformation, and social isolation in contributing to stress and mental morbidity, as documented in several studies [31,33,34].

It has been noticed that when students face high levels of stress or chronic stress, heedless of their age or grade, it can affect their ability to learn, study, memorize, and earn good grades [35]. Therefore, we searched to find the changes that the medical students perceived in studying and learning after the COVID-19 pandemic and found a significant relation among almost all the psychological illnesses studied and perceived learning problems.

The majority of students reported that they had more memorizing problems while learning academic literature (*n* = 216, 54.0%) compared to memorizing them before the pandemic. Most of the study subjects had symptoms of depression and anxiety in the present study as mentioned earlier; therefore, this reduction of memorizing capacity as perceived by students could be understood by the fact revealed in a previous study showing association of depression with an adverse effect on immediate recall of new information along with the amount of acquisition. When compounded by anxiety, it was found that the retrieval of newly learned information became even more difficult [36]. Some other studies [37] also documented that psychosocial stress impaired memory retrieval; this may be reflected by our sample of medical students’ perception that they make more mistakes in solving questions, and their reaction time for solving questions has increased. Another study [38] supports this finding by explaining that when stressful events are separated from learning or are experienced prior to retrieval, such learning or memory is impaired. Most (90.0%) of the medical students thought it was more difficult to study during the COVID- 19 pandemic era in comparison to the pre-pandemic time. A large number of students felt their ability to focus/concentrate on studies had been affected. It was difficult for them to redirect the attention away from the negative, anxiety-provoking cue to their studies. These findings were noted in other studies as well, such as failure to focus oneself–study and preparedness for final year exit exams was noted in a study from Pakistan [19]. Also, a study from New Jersey [39] on the impact of COVID-19 on students found that academic difficulties, such as the ability to focus on academic work, were significantly associated with increased depression, anxiety, somatization, and stress in the post-pandemic period. A recent study with a sample of 1392 veterinary medical students reported that 96.7% of student’s academic performance was affected during the COVID- 19 pandemic lockdown [40].

## 5. Limitation

We do acknowledge some limitations in this study. First, it is an online questionnaire-based survey; we cannot be completely sure that the sample was uniform; maybe students having some psychological symptoms responded more, and this could be the reason for a very high proportion of students showing some degree of assault on their psychological wellbeing. In addition, it was carried out within a limited period of a few months. Therefore, these results might indicate important but short-lived reactions of our participants to the changes. Another limitation was that there is an unequal distribution of responders from different regions of the world as the medical students were only approached through social media groups. Still, our study emphasizes the importance of regional variation in students suffering and indicates a need for well-designed prospective research to enlighten this further. The study was also conducted on a convenience sample that cannot be considered representative of the population of medical students.

## 6. Conclusions

Assault on psychological well-being, brain fog in struggling to memorize, inattention and difficulty concentrating on studies, and perceived overall trouble with learning have emerged as collateral damage from the COVID-19 pandemic to medical students from all over the world. In view of the prediction by Michael Osterholm, the director of the Center for Infectious Disease Research and Policy at the University of Minnesota that a “hurricane is coming.” Describing it as “category 5 or higher,” associated with variant strains of coronavirus [41] our study emphasizes the requirement of a stitch in time to prevent psychological impact and the learning problems that may turn out to be disruptive for our future frontline soldiers serving humanity. This study signposts the tip of the iceberg and forecasts the danger of the detrimental impact of the COVID-19 pandemic and stresses the development of an appropriate contingency plan to mitigate this danger by implementing proper support and counseling to make students more resilient and irrepressible during this unprecedented stress filled situation. The pandemic is now almost entering its third year, but the SARS-CoV-2 virus with its genomic variation continues to circulate throughout communities. This study points the need for a well-planned prospective study with better resources to delineate the perpetrating impairment of this pandemic on the learning and psychological wellbeing of medical students in the future. This may help in understanding and explaining the problem on a massive scale. In addition, undertaking a well-planned clinical trial for stress counseling, management, and resilience training for medical students during this pandemic can also be an eminent step.

## Figures and Tables

**Figure 1 ijerph-18-05792-f001:**
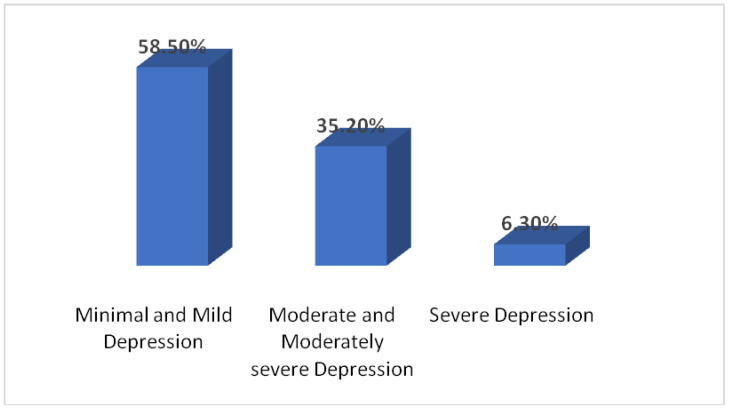
Severity of depression observed in the participants.

**Figure 2 ijerph-18-05792-f002:**
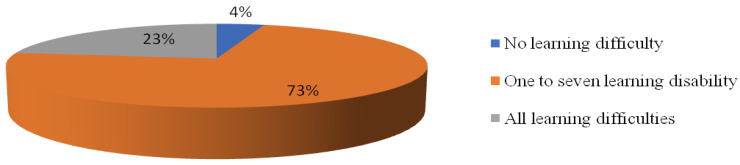
Distribution of a number of learning difficulties present among the participants.

**Table 1 ijerph-18-05792-t001:** Background characteristics of the respondents.

Sr. No.	Variable	Sub-Category	Number	%
1.	Age:	≤25 years	354	84.7
		20–30 years	51	12.2
		>30 years	13	3.1
2.	Gender:	Male	146	34.9
		Female	272	65.1
3.	WHO Regions	African (AFRO)	7	1.7
		Region of America (PAHO)	73	17.4
		South-East Asia Region (SEARO)	112	26.8
		European region (EURO)	23	5.5
		Eastern Mediterranean (EMRO)	194	46.4
		Western Pacific Region (WPRO)	9	2.2
4.	Corona positive cases in the region	<500 cases	109	26.1
		≥500 cases	309	73.9
5.	Corona mortality in the region	<100 cases	116	38.5
		≥100 cases	257	61.5
6.	Level of study	Undergraduate medical student	292	69.9
		Postgraduate medical students	126	30.1
7.	Employment:	Yes employed	102	24.4
		Not employed	316	75.6
8.	Financial support	Government	48	11.5
		Parents	312	74.6
		Self	58	13.9
9.	Study enrolment:	Enrolled with institute	234	56.0
		Self-study	184	44.0
10.	Study environment:	Home	365	87.3
		Any other	53	12.7
11.	History of previous psychiatric illness	Yes	71	17.0
		No	347	83.0

**Table 2 ijerph-18-05792-t002:** Prevalence of psychiatric illness in the subjects.

Sr.No.	Disorder	Yes	No
1.	Depression	386 (92.3%)	32 (7.7%)
2.	Anxiety (GAD)	158 (37.8%)	260 (62.2%)
3.	Obsessive-Compulsive Disorder (OCD)	225 (53.8%)	193 (46.2%)
4.	Post-Traumatic Stress Disorder (PTSD)	129 (30.9%)	289 (69.1%)

**Table 3 ijerph-18-05792-t003:** Relation between previous history of psychiatric illness and the prevalence of different psychiatric illness in the present pandemic era.

Sr.No.	Psychiatric Illness in the Present Pandemic Era		Previous History of Psychiatric Illness		*p*-Value
			Yes Number & %	No Number & %	
1.	Depression	Yes	71 (100%)	Zero (0%)	0.008
		No	315 (90.8%)	32 (9.2%)	
2.	Anxiety	Yes	44 (62.0%)	27 (38.0%)	0.001
		No	114 (32.9%)	233 (67.1%)	
3.	OCD	Yes	46 (64.8%)	25 (35.2%)	0.042
		No	179 (51.6%)	168 (48.4%)	
4.	PTSD	Yes	34 (47.9%)	37 (52.1%)	0.001
		No	95 (27.4%)	252 (72.6%)	

**Table 4 ijerph-18-05792-t004:** Relation of Psychiatric Illness with background characteristics.

Variables	Yes Number (%)	No Number (%)	*p*-Value	x^2^
Depression *n* = 386 *				
Gender				
Male	127 (32.9%)	19 (59.4%)	0.003	9.4
Female	259 (67.1%)	13 (40.6%)		
Region *				
AFRO	7 (1.8%)	0 (0.0%)	0.032	12.24
PAHO	70 (18.1%)	3 (9.4%)		
SEARO	107 (27.7%)	5 (15.6%)		
EURO	23 (6.0%)	0 (0.00%)		
EMRO	170 (44.1%)	24 (75.0%)		
WPRO	9 (2.3%)	0 (0.00%)		
Corona Mortality				
<100	155 (40.2%)	6 (18.8%)	0.017	5.71
≥100	231 (59.8%)	26 (81.2%)		
Employment				
Employed	86 (22.3%)	16 (50%)	<0.001	12.31
Not Employed	300 (77.7%)	16 (50%)		
History of Previous Psychiatric Illness				
Yes	71 (18.4%)	0 (0.0%)	0.008	7.09
No	315 (81.6%)	32 (100%)		
Anxiety (GAD) *n* = 158 *				
Region *				
AFRO	0 (0.0%)	7 (2.7%)	0.04	11.61
PAHO	33 (20.9%)	40 (15.4%)		
SEARO	33 (20.9%)	79 (30.4%)		
EURO	12 (7.6%)	11 (4.2%)		
EMRO	76 (48.1%)	118 (45.4%)		
WPRO	4 (2.5%)	5 (1.9%)		
History of Previous Psychiatric Illness				
Yes	44 (27.8%)	27 (10.4%)	<0.001	21.25
No	114 (72.2%)	233 (88.6%)		
Obsessive Compulsive Disorder (OCD) *n* =225 *				
Region *				
AFRO	1 (0.4%)	6 (3.1%)	0.01	13.44
PAHO	49 (21.8%)	24 (12.4%)		
SEARO	56 (24.9%)	56 (29.0%)		
EURO	10 (4.5%)	13 (6.8%)		
EMRO	102 (45.3%)	92 (47.7%)		
WPRO	7 (3.1%)	2 (1.0%)		
History of Previous Psychiatric Illness				
Yes	46 (20.4%)	25 (13.0%)	0.04	4.13
No	179 (79.6%)	168 (87.0%)		
Post-Traumatic Stress Disorder (PTSD)*n* =129 *				
Gender				
Male	33 (25.6%)	113 (39.1%)	0.007	11.62
Female	96 (74.4%)	176 (60.9%)		
History of Previous Psychiatric Illness				
Yes	34 (26.4%)	37 (12.8%)	0.001	11.62
No	95 (73.6%)	252 (87.2%)		

* = number of students who had the symptoms.

**Table 5 ijerph-18-05792-t005:** Prevalence of learning difficulties as perceived by the students.

Sr.No.	Learning Difficulty Questions	Yes Number &Percentage	No Number &Percentage
1.	Does the corona pandemic have an effect on your memorizing capacity	216 (54.0%)	184(46.0%)
2.	Do you think your ability to focus/concentrate on studies is affected?	268 (67.0%)	132 (33.0%)
3.	Have you noticed the reaction time for solving questions has increased?	222 (55.5%)	178(44.5%)
4.	Are you making more mistakes in solving questions?	214 (53.5%)	168 (46.5%)
5.	Has your usual timing of study changed from pre-corona pandemic time?	255 (63.8%)	145 (36.2%)
6.	Is it difficult to redirect the attention away from the negative, anxiety-provoking cue to your studies?	250 (62.5%)	150 (37.5%)
7.	Are you unable to adapt to the presently changing environment?	224 (56.0%)	176 (44.0%)
8.	Do you think it is more difficult to study during the COVID- 19 pandemic in comparison to the pre-pandemic time?	360 (90.0%)	40 (10.0%)

The total number of students = 400 as 18 students did not perceive greater learning difficulty than in the pre-pandemic era.

**Table 6 ijerph-18-05792-t006:** Multiple binary logistic regression analysis of student diagnosed with any psychiatric disorder-related factors.

Variable	Coefficient	Standard Error	*p*-Value	Odds Ratio	95% Confidence Interval
Age	−0.0743	0.3495	0.8317	0.9284	(0.4680, 1.8419)
Gender	0.2094	0.3108	0.5004	1.2329	(0.6705, 2.2671)
Country	0.0827	0.1195	0.4889	1.0862	(0.8594, 1.3728)
Level of Study	0.1876	0.3507	0.5926	1.2064	(0.6067, 2.3987)
Study Enrollment	0.1595	0.2950	0.5887	1.1729	(0.6579, 2.0912)
Study Environment	0.1334	0.4097	0.7447	1.1427	(0.5120, 2.5506)
Corona Positive	−0.7662	0.4900	0.1179	0.4648	(0.1779, 1.2143)
Corona Mortality	1.0117	0.4494	0.0244	2.7503	(1.1398, 6.6361)
GAD	0.9171	0.3096	0.0031	2.5020	(1.3638, 4.5900)
PQ	20.4920	7468.5655	0.9978	03733.0052	(0.0000, 0.0000)
OCD	−0.2297	0.3488	0.5101	0.7948	(0.4012, 1.5744)
PTSD	0.4554	0.3514	0.1950	1.5768	(0.7919, 3.1395)
Constant	−24.0195	7468.5656	0.9974		

## Data Availability

Not applicable.

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
