# Peer review of "COVID-19 Pandemic Affects the Medical Students’ Learning Process and Assaults Their Psychological Wellbeing"

_ijerph, 2021, doi:10.3390/ijerph18115792_

Round 1
Reviewer 1 Report
I have read this manuscript with great pleasure. The subject seems interesting to me. I am totally sure that the pandemic we are experiencing is psychologically affecting our entire society, including, of course, all students, both in medicine and in other disciplines.
In this manuscript, a cross-sectional description is made to see how the restrictions imposed to try to control the pandemic are affecting medical students and how they affect their learning process.
Unfortunately, I must say that the manuscript is seriously flawed. I think the investigation was not done correctly. The methodological design is too poor and I think that many biases have not been taken into account.
The first thing that caught my attention is that the study is carried out worldwide. Usually this type of study is done in smaller geographic areas. In all these countries the native language is not English. Therefore, here we already have a significant bias. The measurement questionnaires must be previously translated and subsequently validated for the specific population to which they are directed.
The second thing that struck me was the size of the sample. To calculate the sample size, it has been assumed that the total population of medical students in the world is 5,000. Doesn't this figure seem too low to you? Currently, in Spain alone we have 42,000 registered medical students. therefore, the chosen sample seems insignificant to me.
The inclusion criteria also lead me to question bias. The only condition to participate is to be a medical student regardless of other relevant factors such as age, having a previous mental health disorder, not controlling the language, having socioeconomic problems. It seems that with being a medical student anything goes ...
on the other hand the methodology is very poor. The study is only limited to making contingency tables to describe categorical variables. At no time is a quantitative analysis done. Not even a comparative analysis of the different study variables is done. Nor is a multivariate study carried out to see the dependence of some variables with others.
I could list an endless list of flaws. . Unfortunately, I believe that this manuscript does not reach the minimum of scientific rigor to be published in a journal of this category.
Thanks
Author Response
Reviewer-The first thing that caught my attention is that the study is carried out worldwide. Usually this type of study is done in smaller geographic areas. In all these countries the native language is not English. Therefore, here we already have a significant bias. The measurement questionnaires must be previously translated and subsequently validated for the specific population to which they are directed.
Answer-The language was not included in the inclusion criteria as the study participants were from social media groups already communicating in English language.
Reviewer-The second thing that struck me was the size of the sample. To calculate the sample size, it has been assumed that the total population of medical students in the world is 5,000. Doesn't this figure seem too low to you? Currently, in Spain alone we have 42,000 registered medical students. therefore, the chosen sample seems insignificant to me.
Answer-The total population of approachable Medical students may be 5000” : this was written by typographical mistake actually our aim was to take 20,000 approachable population as according to the Roasoft online sample size calculator “The sample size doesn't change much for populations larger than 20,000”
Reviewer-The inclusion criteria also lead me to question bias. The only condition to participate is to be a medical student regardless of other relevant factors such as age, having a previous mental health disorder, not controlling the language, having socioeconomic problems. It seems that with being a medical student anything goes ...
Answer-Revised Inclusion Criteria: The study included;
- Undergraduate medical students in the clinical phase of their studies
- Postgraduate students who are preparing actively and planning to appear in any qualification equivalence exams like PLAPS, USMLE or those preparing for higher qualifications like Board exams, membership or fellowship exams.
There is no restriction for age or months after graduation as Medical studies can be at any age and can be planned any time after graduation. The language was not included in the inclusion criteria as the study participants were from social media groups already communicating in English language. Previous psychiatric illness and socioeconomic aspect has been covered in the questions addressing the background characterstics.
Reviewer-on the other hand the methodology is very poor. The study is only limited to making contingency tables to describe categorical variables. At no time is a quantitative analysis done. Not even a comparative analysis of the different study variables is done. Nor is a multivariate study carried out to see the dependence of some variables with others.
Answer The analysis is improved by adding further multivariate study has been included as suggested.
Reviewer 2 Report
This study evaluated the impact of the COVID-19 pandemic in terms of psychological well being and learning difficulties experienced by medical studients. The study was conducted in a sample of 418 undergraduate and postgraduate students from different countries. The topic is relevant and results are of interest. I have some suggestions on aspects that might be improved.
While the authors calculated sample size for the study, the choice to include respondents from all over the world led to include a small number of participants from specific regions. On one hand this choice allowed to have some information on different geographic areas, on the other hand some areas only include a small number of participants.
- How was the number of 5000 approachable medical students defined?
- I think the authors should better specify inclusion criteria for responding to the survey (at present they report "All the students studying medicine"). Since 44% of participants were not enrolled at any institute, please define which participants were eligilble to be included (for instance X months after graduation)
- At page 6, the authors report: "Among 389 students, who had a psychiatric illness ..". I think the authors should refer to symptoms rather than imply that a diagnosis of psychiatric illness was made based on the answers
- Rather than just using chi-square test, why not conduct multivariate analysis to assess the contribution of multiple variables jointly?
- The authors acknowledge as a limitation that fact that people with a higher psychological distress might be more likely to respond to the survey. I think this aspect should be stressed more, and that the authors should write that the study was conducted on a convenience sample which cannot be considered to be representative of the population of medical students.
- Throughout the text, several words are not separated by white spaces e differnt fonts are used
Author Response
Reviewer- How was the number of 5000 approachable medical students defined?
Answer-
“The total population of approachable Medical students may be 5000” : this was written by typographical mistake actually our aim was to take 20,000 approachable population as according to the Roasoft online sample size calculator “The sample size doesn't change much for populations larger than 20,000”
(Reference: http://www.raosoft.com/samplesize.html). The correction is done in the methodology section
Reviewer-I think the authors should better specify inclusion criteria for responding to the survey (at present they report "All the students studying medicine"). Since 44% of participants were not enrolled at any institute, please define which participants were eligilble to be included (for instance X months after graduation)
Answer- Revised Inclusion Criteria: The study included;
- Undergraduate medical students in the clinical phase of their studies
- Postgraduate students who are preparing actively and planning to appear in any qualification equivalence exams like PLAPS, USMLE or those preparing for higher qualifications like Board exams, membership or fellowship exams.
There is no restriction for age or months after graduation as Medical studies can be at any age and can be planned any time after graduation. The language was not included in the inclusion criteria as the study participants were from social media groups already communicating in English language. Previous psychiatric illness and socioeconomic aspect has been covered in the questions addressing the background characterstics.
Reviewer- - At page 6, the authors report: "Among 389 students, who had a psychiatric illness ..". I think the authors should refer to symptoms rather than imply that a diagnosis of psychiatric illness was made based on the answers
Answer-As suggested by the respected reviewer, this has been included.
Reviewer-Rather than just using chi-square test, why not conduct multivariate analysis to assess the contribution of multiple variables jointly?
Answer- The analysis is improved by adding further multivariate study has been included as suggested.
Reviewer- The authors acknowledge as a limitation that fact that people with a higher psychological distress might be more likely to respond to the survey. I think this aspect should be stressed more, and that the authors should write that the study was conducted on a convenience sample which cannot be considered to be representative of the population of medical students.
Answer-As suggested by the respected reviewer, this has been included.
Reviewer- Throughout the text, several words are not separated by white spaces e different fonts are used
Answer-As suggested by the respected reviewer, this has been corrected.
Round 2
Reviewer 1 Report
In this version of the manuscript, the quality has been improved somewhat, following most of the advice provided by the reviewers.
It is true that the article has great limitations, which are recognized by the authors. If similar studies are compared in other countries, the results are similar.
With these considerations in mind, I think the article could be published with the editor's approval.
Thank you
Kind regards
Author Response
I sincerely thank the respected reviewer for considering our reply and the positive response.
Reviewer 2 Report
The authors addressed most comments.
- At page 4, the following sentence "A multivariate logistic regression analysis had been used tosee the dependence of some variables on others" should be revised. Authors should clearly indicate the outcome of the regression model and the included variables.
- When reporting the results from the logistic regression model at page 9, I think the authors should also mention that the "Corona mortality" variable was associated with the outcome.
- Please reread the manuscript carefully, as there are still many words attached together or other typos throughout the text.
Author Response
Reviewer: At page 4, the following sentence "A multivariate logistic regression analysis had been used to see the dependence of some variables on others" should be revised. Authors should clearly indicate the outcome of the regression model and the included variables.
Response: Multivariate analysis was performed using the binary logistic regression model to determine the OR for student diagnosed with any psychiatric disorder and related factors such as age, gender, country, level of study, study enrollment, study environment, corona positive, corona mortality, GAD, PHQ,OCD and PTSD. (please see page 4 para 3 highlighted in yellow)
Reviewer: When reporting the results from the logistic regression model at page 9, I think the authors should also mention that the "Corona mortality" variable was associated with the outcome.
Response: The multivariate analysis revealed that, GAD-7 scales for anxiety screening and corona mortality was associated with a diagnosis of psychiatric disorder, as students from GAD-7 anxiety scales and corona mortality had a 2.50-fold and 2.57-fold greater chance of exhibiting psychiatric disorder respectively. ( Please see Page 10 para 1 highlighted in yellow)
Reviewer: Please reread the manuscript carefully, as there are still many words attached together or other typos throughout the text.
Response: As suggested by the respected reviewer we have removed the typos.